# Analysis of Deep Convolutional Neural Networks Using Tensor Kernels and Matrix-Based Entropy

**DOI:** 10.3390/e25060899

**Published:** 2023-06-03

**Authors:** Kristoffer K. Wickstrøm, Sigurd Løkse, Michael C. Kampffmeyer, Shujian Yu, José C. Príncipe, Robert Jenssen

**Affiliations:** 1Machine Learning Group, Department of Physics and Technology, UiT The Arctic University of Norway, NO-9037 Tromsø, Norway; sigurd.lokse@uit.no (S.L.); michael.c.kampffmeyer@uit.no (M.C.K.); s.yu3@vu.nl (S.Y.); robert.jenssen@uit.no (R.J.); 2Norwegian Computing Center, Department of Statistical Analysis and Machine Learning, 114 Blindern, NO-0314 Oslo, Norway; 3Computational NeuroEngineering Laboratory, Department of Electrical and Computer Engineering, University of Florida, Gainesville, FL 32611, USA; principe@cnel.ufl.edu; 4Department of Computer Science, Vrije Universiteit Amsterdam, 1081 HV Amsterdam, The Netherlands; 5Department of Computer Science, University of Copenhagen, Universitetsparken 1, 2100 Copenhagen, Denmark

**Keywords:** information theory, deep learning, information plane, kernels methods

## Abstract

Analyzing deep neural networks (DNNs) via information plane (IP) theory has gained tremendous attention recently to gain insight into, among others, DNNs’ generalization ability. However, it is by no means obvious how to estimate the mutual information (MI) between each hidden layer and the input/desired output to construct the IP. For instance, hidden layers with many neurons require MI estimators with robustness toward the high dimensionality associated with such layers. MI estimators should also be able to handle convolutional layers while at the same time being computationally tractable to scale to large networks. Existing IP methods have not been able to study truly deep convolutional neural networks (CNNs). We propose an IP analysis using the new matrix-based Rényi’s entropy coupled with tensor kernels, leveraging the power of kernel methods to represent properties of the probability distribution independently of the dimensionality of the data. Our results shed new light on previous studies concerning small-scale DNNs using a completely new approach. We provide a comprehensive IP analysis of large-scale CNNs, investigating the different training phases and providing new insights into the training dynamics of large-scale neural networks.

## 1. Introduction

Although deep neural networks (DNNs) are at the core of most state-of-the art systems in computer vision, the theoretical understanding of such networks is still not at a satisfactory level [1]. In order to provide insight into the inner workings of DNNs, the prospect of utilizing the mutual information (MI), a measure of dependency between two random variables, has recently garnered a significant amount of attention [1,2,3,4,5,6,7,8]. Given the input variable *X* and the desired output *Y* for a supervised learning task, a DNN is viewed as a transformation of *X* into a representation that is favorable for obtaining a good prediction of *Y*. By treating the output of each hidden layer as a random variable *T*, one can model the MI I(X;T) between *X* and *T*. Likewise, the MI I(T;Y) between *T* and *Y* can be modeled. The quantities I(X;T) and I(T;Y) span what is referred to as the information plane (IP). Several works have unveiled interesting properties of the training dynamics through IP analysis of DNNs [4,7,9,10,11]. Figure 1, produced using our proposed measure, illustrates one such insight that is similar to the observations of [1], where training can be separated into two distinct phases: the fitting phase and the compression phase.

The claim of a fitting and compression phase has been highly debated as subsequent research has linked the compression phase to the saturation of neurons [5] or clustering of the hidden representations [9]. Recent studies [1,5,7,9] have been focused on small-scale networks or non-convolutional networks, as the current MI estimators cannot tackle the tensor representations produced by convolutional layers or do not scale well to convolutional layers with many filters [6]. This severely limits the scope of IP analysis, as most real-world applications rely on large-scale CNNs. In this work, we continue the IP line of research through a new matrix-based entropy functional [6,7,12]. We provide insights on this functional by linking the functional to well-understood measures from the kernel literature and propose a new kernel tensor-based approach to the matrix-based entropy functional. Furthermore, we give a new formulation of the matrix-based entropy that is closely connected to measures from quantum information theory. Using the proposed estimator, we provide an analysis of large-scale DNNs and give a new information theoretic understanding of the training procedure of DNNs. Using the proposed measure, we investigate the claim of [3] that the entropy H(X)≈I(T;X) and H(Y)≈I(T;Y) in high dimensions (in which case MI-based analysis would be meaningless). The contributions of this work can be summarized as:We propose a kernel tensor-based approach to the matrix-based entropy functional that is designed for measuring MI in large-scale convolutional neural networks (CNNs).We provide new insights on the matrix-based entropy functional by showing its connection to well-known quantities in the kernel literature such as the kernel mean embedding and maximum mean discrepancy. Furthermore, we show that the matrix-based entropy functional is closely linked with the von Neuman entropy from quantum information theory.Our results indicate that the compression phase is apparent mostly for the training data and less so for the test data, particularly for more challenging datasets. When using a technique such as early stopping to avoid overfitting, training tends to stop before the compression phase occurs (see Figure 1).

## 2. Related Work

Analyzing DNNs in the IP was first proposed by [13] and later demonstrated by [1]. Among other results, the authors studied the evolution of the IP during the training process of DNNs and noted that the process was composed of two different phases. First, there is an initial fitting phase where I(T;Y) increases, which is followed by a phase of compression where I(X;T) decreases. These results were later questioned by [5], who argued that the compression phase is not a general property of the DNN training process but rather an effect of different activation functions. However, a recent study by [4] seems to support the claim of a compression phase regardless of the activation function. The authors argue that the base estimator of MI utilized by [5] might not be accurate enough and demonstrate that a compression phase does occur, but the amount of compression can vary between different activation functions. Another recent study by [10] also reported a compression phase but highlighted the importance of adaptive MI estimators. They also showed that when L2 regularization was included in the training, compression was observed regardless of the activation function. In addition, some recent studies have discussed the limitations of the IP framework for analysis and optimization for particular types of DNN [14,15]. Furthermore, ref. [16] investigated similarities between hidden layers and between hidden layers of different networks, but they did so only for the representation obtained after the networks were fully trained. The dynamics of large-scale DNNs was investigated [17] using MINE [18]. A number of information plane-related studies have also been discussed in [19].

On a different note, Ref. [3] proposed an evaluation framework for DNNs based on the IP and demonstrated that MI can be used to infer the capability of DNNs to recognize objects for an image classification task. Furthermore, the authors argue that when the number of neurons in a hidden layer grows large, I(T;X) and I(Y;T) barely change and are, using [3] terminology, approximately deterministic, i.e., I(T;X)≈H(X) and I(T;Y)≈H(Y). Therefore, they only model the MI between *X* and the last hidden layer—that is, the output of the network—and the last hidden layer and *Y*.

Ref. [7] investigated a matrix-based measure of MI for analyzing different data processing inequalities in feed-forward stacked autoencoders (SAEs), concluding that the compression phase in the IP of SAEs is determined by the values of the SAE bottleneck layer size and the intrinsic dimensionality of the given data. In follow-up work, a simplistic analysis of small convolutional neural networks (CNNs) was provided in [6]; however, it was based on a multivariate extension [20] of matrix-based Renyi entropy that does not scale well numerically or computationally in the number of feature maps. The interested reader can find additional information on the multivariate extension by [20] in Appendix A.

Recently, the information plane of quantized neural networks was modeled [8], which allowed for an exact analysis of its dynamics. In addition, log-determinant entropy has been used for information plane analysis [11]. Lastly, information plane analysis has also been used to improve the understanding of graph convolutional neural networks [21].

In this paper, we continue the recent trend of leveraging the new matrix-based measures of entropy. We contribute both new insight to the definition of these measures (see Section 3.2.1), and importantly, we extend matrix-based measures of entropy to exploit tensor kernels to enable the first IP analysis of large-scale CNNs by treating feature maps as tensors (Section 3.3).

## 3. Materials and Methods

### 3.1. Preliminaries on Matrix-Based Information Measures

For the benefit of the reader, we review in this section first the theory underlying the recent matrix-based measures of entropy and mutual information. Thereafter, we contribute a new special case interpretation of the definition of matrix-based Renyi entropy (Section 3.2.1) and give new insights on the link to other measures in the kernel literature before presenting our new tensor-based approach.

#### 3.1.1. Matrix-Based Entropy and Mutual Information

The matrix-based measure of entropy, originally proposed by [12], is built on kernel matrices obtained from raw data, involving no explicit density estimation or binning procedure:

**Definition 1** ([12])**.**
*Let $x_{i}\in X,\;i=1,2,\ldots ,N$ denote data points and let κ:X×X↦R be an infinitely divisible positive definite kernel [22]. Given the kernel matrix $K\in R^{N\times N}$ with elements ${\left(K\right)}_{ij}=\kappa \left(x_{i},x_{j}\right)$ and the matrix $A,\;{\left(A\right)}_{ij}=\frac{1}{N}\frac{{\left(K\right)}_{ij}}{\sqrt{{\left(K\right)}_{ii}{\left(K\right)}_{jj}}}$, the matrix-based Rényi’s α-order entropy is given by*
(1)$$\begin{array}{cc} S_{\alpha }\left(A\right) & =\frac{1}{1-\alpha }\log_{2}\left(\mathrm{tr}(A^{\alpha })\right)\\ \; & =\frac{1}{1-\alpha }\log_{2}\left[\sum_{i=1}^{N}\lambda_{i}{\left(A\right)}^{\alpha }\right]. \end{array}$$

Here, $\lambda_{i}\left(A\right)$ denotes the ith eigenvalue of the matrix A. Equation (Equation 1) is a measure of an entropy-like quantity that satisfies Renyi’s axiomatic characterization of entropy [23], which is referred to as matrix-based Renyi entropy. In addition to the matrix-based entropy, ref. [12] also defined the matrix-based joint entropy between x∈X and y∈Y as
(2)$$S_{\alpha }\left(A_{X},A_{Y}\right)=S_{\alpha }\left(\frac{A_{X}\circ A_{Y}}{\mathrm{tr}(A_{X}\circ A_{Y})}\right),$$
where $x_{i}$ and $y_{i}$ are two different representations of the same object and ∘ denotes the Hadamard product. Finally, the MI is, similar to Shannon’s formulation, defined as
(3)$$I_{\alpha }\left(A_{X};A_{Y}\right)=S_{\alpha }\left(A_{X}\right)+S_{\alpha }\left(A_{Y}\right)-S_{\alpha }\left(A_{X},A_{Y}\right).$$

The properties of these quantities were analyzed in detail in [12], but we want to highlight some important properties and provide links with other measures in the kernel literature.

Information theoretic measures are developed in such a way that they satisfy certain axioms. The measures presented above satisfy the axioms put forth by [23] and are closely connected with quantum information theory [24]. In quantum statistical mechanics, the Von Neumann’s entropy [24] is defined as
(4)S(ρ)=−tr(ρlogρ),
where ρ is a density matrix that described a quantum mechanical system. If ρ is written in terms of its eigenvalues, $\lambda_{i}\left(\rho \right)$, then Equation (Equation 4) can also be formulated as
(5)$$S\left(\rho \right)=-\sum_{i=1}^{N}\lambda_{i}\left(\rho \right)\log\left[\lambda_{i}\left(\rho \right)\right].$$

In addition, the quantum extensions of Renyi’s entropy [25] that is defined as
(6)$$S_{\alpha }\left(\rho \right)=\frac{1}{1-\alpha }\log\left[\mathrm{tr}\left(\rho^{\alpha }\right)\right],$$
bears a close resemblance to the matrix-based definition of entropy in Equation (Equation 1). While some properties of Equations (Equation 4) and (Equation 6) can also be extended to Equation (Equation 1), it is important to note that the two approaches are very different since the matrix-based framework is built around kernel matrices obtained directly from raw data.

#### 3.1.2. Bound on Matrix-Based Entropy Measure

Not all measures of entropy have the same properties. Many of the estimators used and developed for Shannon suffer from the curse of dimensionality [26]. In contrast, Renyi’s entropy measures have the same functional form of the statistical quantity in a reproducing kernel Hilbert space (RKHS), thus capturing properties of the data population. Essentially, we are projecting marginal distribution to an RKHS in order to measure entropy and MI. This is similar to the approach of maximum mean discrepancy and the kernel mean embedding [27,28]. The connection with the data population can be shown via the theory of covariance operators. The covariance operator G:H→H is defined through the bilinear form
(7)$$\begin{array}{cc} G\left(f,g\right)=\langle f,Gg\rangle & =\int_{X}\langle f,\psi (x)\rangle \langle \psi (x),g\rangle dP_{X}\left(x\right)\\ \; & =E_{X}\left\{f(X),g(Y)\right\} \end{array}$$
where $P_{X}$ is a probability measure and f,g∈H. Based on the empirical distribution $P_{N}=\frac{1}{N}\sum_{i=1}^{N}\delta_{x_{i}}\left(x\right)$, the empirical version $\hat{G}$ of *G* obtained from a sample $x_{i}$ of size *N* is given by:(8)$$\begin{array}{cc} \langle f,{\hat{G}}_{Ng}\rangle =\hat{G}\left(f,g\right) & =\int_{X}\langle f,\psi (x)\rangle \langle \psi (x),g\rangle dP_{X}\left(x\right)\\ \; & =\frac{1}{N}\sum_{i=1}^{N}\langle f,\psi (x_{i})\rangle \langle \psi (x_{i}),g\rangle \end{array}$$

By analyzing the spectrum of $\hat{G}$ and *G*, Ref. [12] showed that the difference between tr(G) and $\mathrm{tr}(\hat{G})$ can be bounded, as stated in the following proposition:

**Proposition 2.** 
*Let $P_{N}=\frac{1}{N}\sum_{i=1}^{N}\delta_{x_{i}}\left(x\right)$ be the empirical distribution. Then, as a consequence of Proposition 6.1 in [12], $tr\left[{\hat{G}}_{N}^{\alpha }\right]=tr\left[{\left(\frac{1}{N}K\right)}^{\alpha }\right]$. The difference between tr(G) and $\mathrm{tr}(\hat{G})$ can be bounded under the conditions of Theorem 6.2 in [12] and for α>1, with probability 1-δ*

(9)
$$\left|tr\left(G^{\alpha }\right)-tr\left({\hat{G}}_{N}^{\alpha }\right)\right|\leq \alpha C\sqrt{\frac{2\log\frac{2}{\delta }}{N}}$$

*where C is a compact self-adjoint operator.*


### 3.2. Analysis of Matrix-Based Information Measures

In this section, we present new theoretical insights into the Matrix-Based Information Measures.

#### 3.2.1. A New Special-Case Interpretation of the Matrix-Based Renyi Entropy Definition

In previous works, the α in Equation (Equation 1) has been ad hoc set to a value of 1.01 in order to approximate Shannon’s entropy [6,7]. For α=1, both the denominator and the numerator become zero, so Equation (Equation 1) cannot be used directly in this case. However, as a contribution to the matrix-based Renyi entropy theory, we show here that for the case α→1, Equation (Equation 1) can be expressed similarly to the matrix-based Von Neumann’s entropy [24], resembling Shannon’s definition over probability states and expressed as
(10)$$\lim_{\alpha \to 1}S_{\alpha }\left(A\right)=-\sum_{i=1}^{N}\lambda_{i}\left(A\right)\log_{2}\left[\lambda_{i}\left(A\right)\right].$$
Equation (Equation 1) can be proved using L’Hôpital’s rule as follows:

**Proof.** (11)$$\lim_{\alpha \to 1}S_{\alpha }\left(A\right)=\lim_{\alpha \to 1}\frac{1}{1-\alpha }\log_{2}\left(\sum_{i=1}^{n}\lambda_{i}^{\alpha }\right)\to \frac{0}{0},$$
since $\sum_{i=1}^{N}\lambda_{i}=\mathrm{tr}\left(A\right)=1$. L’Hôpital’s rule yields
(12)$$\begin{array}{cc} \lim_{\alpha \to 1}S_{\alpha }\left(A\right) & =\lim_{\alpha \to 1}\frac{\frac{\partial }{\partial \alpha }\log_{2}\left[\sum_{i=1}^{n}\lambda_{i}{\left(A\right)}^{\alpha }\right]}{\frac{\partial }{\partial \alpha }\left(1-\alpha \right)}\\ \; & =-\frac{1}{\ln2}\lim_{\alpha \to 1}\frac{\sum_{i=1}^{n}\lambda_{i}{\left(A\right)}^{\alpha }\ln\left[\lambda_{i}\left(A\right)\right]}{|\sum_{i=1}^{n}\lambda_{i}{\left(A\right)}^{\alpha }|}\\ \; & =-\sum_{i=1}^{n}\lambda_{i}\left(A\right)\log_{2}\left[\lambda_{i}\left(A\right)\right]. \end{array}$$□

#### 3.2.2. Link to Measures in Kernel Literature and Validation on High-Dimensional Synthetic Data

An interesting aspect of the matrix-based measure of entropy is the special case connection with the theory of maximum mean discrepancy and Hilbert–Schmidt norms via covariance operators [29]. Let *G* be the covariance operator (see [28] for details), then, for the particular case of α=2, the empirical trace of the covariance operator, $\mathrm{tr}(G^{2})$, is given by $\mathrm{tr}(A^{2})$. Furthermore,
(13)$$\begin{array}{cc} \mathrm{tr}(G^{2})\; & =\langle \int_{X}\kappa \left(\cdot ,x\right)dP_{X}\left(x\right),\int_{X}\kappa \left(\cdot ,y\right)dP_{X}\left(y\right),\rangle \\ \; & =\left|\right|\mu_{X}{\left|\right|}_{K}^{2}, \end{array}$$
where $\mu_{X}=\int_{X}\kappa \left(\cdot ,x\right)dP_{X}\left(x\right)$ is the *kernel mean map* [30], i.e., an embedding of the probability measure $P_{X}\left(x\right)$ in a reproducing kernel Hilbert space (RKHS). Thus, the matrix A can be related to an empirical covariance operator on embeddings of probability distributions in an RKHS. Moreover, ref. [12] showed that under certain conditions, Equation (Equation 1) converges to the trace of the underlying covariance operator, as shown in Proposition 2 in Section 3.1.2. Notice that the dimensionality of the data does not appear in Proposition 2. This means that $S_{\alpha }\left(A\right)$ captures properties of the distribution with a certain robustness with respect to high-dimensional data. This is a beneficial property compared to KNN and KDE-based information measures used in previous works [5,10], which have difficulties handling high-dimensional data [26]. Some measures of entropy developed for measuring the Shannon entropy suffer from the curse of dimensionality [26]. In addition, there is no need for any binning procedure utilized in previous works [1], which are known to struggle with the ReLU activation function commonly used in DNNs [5]. While Equation (Equation 13) is not explicitly used in the remainder of our manuscript, we believe that these insights provides a deeper understanding of the inner workings of the matrix-based entropy measure.

To examine the behavior of the matrix-based measures described in Section 3.1, we have conducted a simple experiment on measuring entropy and mutual information in high-dimensional data following a normal distribution with known mean and covariance matrix. In the particular case of the normal distribution, the entropy and mutual information can be calculated analytically. The entropy can be calculated as:(14)$$H\left(N_{0}\right)=\frac{1}{2}\log\left({\left(2\pi e\right)}^{d}\det\left(\Sigma_{0}\right)\right),$$
where $N_{0}$ denotes a normal distribution with mean vector $\mu_{0}$ covariance matrix $\Sigma_{0}$, and dimensionality *d*. For mutual information, we use the experimental setup considered in [31]. Let *Z* have a d+1 dimensional Gaussian distribution with covariance matrix $\Sigma_{z}$. Next, let $X=(Z_{1},\ldots ,Z_{d})$ and $Y=Z_{d+1}$. Then, their mutual information satisfies:(15)$$\begin{array}{cc} I(X,Y) & =I\left(Z_{1},\ldots ,Z_{d+1}\right)-I\left(Z_{1},\ldots ,Z_{d}\right) \end{array}$$(16)$$\begin{array}{cc} & =-\frac{1}{2}\log\left(\frac{\det(\Sigma_{z})}{\det(\Sigma_{x})}\right), \end{array}$$
where $\Sigma_{x}$ is the covariance matrix of *X*. We consider five cases of (d+1)-dimensional Gaussian distributions with mean zero, unit variance, and an increasing amount of dependence. The unit variance covariance matrices for the five cases are given as follows:

$$\left\{\begin{array}{cc} \Sigma_{A}\left(i,j\right)=0.1, & \mathrm{for}\;i\neq j,\\ \Sigma_{B}\left(i,j\right)=0.25, & \mathrm{for}\;i\neq j,\\ \Sigma_{C}\left(i,j\right)=0.5, & \mathrm{for}\;i\neq j,\\ \Sigma_{D}\left(i,j\right)=0.75, & \mathrm{for}\;i\neq j,\\ \Sigma_{E}\left(i,j\right)=0.9, & \mathrm{for}\;i\neq j. \end{array}\right.$$
The left plot in Figure 2 displays the entropy of a 100-dimensional normal distribution with zero mean and isotropic covariance matrix, which is calculated using Equation (Equation 14). The right plot in Figure 2 displays the estimated entropy using Equation (Equation 1) on 500 randomly drawn samples from $N_{0}$, which are computed on all 500 samples and in batches of 100. The results show how the estimated entropy follows the same trend as the analytically computed entropy. We quantitativly evaluate the correlation between the analytic quantity and the estimated quantity by calculating Pearson’s correlation coefficient and find that that both the full data and batch-wise estimation are highly correlated with the analytic calculation (correlation ≈0.99, *p*-value ≤0.01). For mutual information, we generate 500 samples from a 100 + 1 dimensional Gaussian distribution and compare the exact and estimated mutual information, which are both based on all 500 samples and in the batch-wise setting. The left plot of Figure 3 shows the mutual information between *X* and *Y* calculated using Equation (16) for the five cases described above. The right part of Figure 3 shows the estimated mutual information using Equation (Equation 3), which is computed on all 500 samples and in batches of 100. Again, the result shows how the estimated values follow the same trend as the exact mutual information values. Similarly as with the entropy, we quantitativly evaluate the correlation between the analytic quantity and the estimated quantity by calculating Pearson’s correlation coefficient and find that that both the full data and batch-wise estimation are highly correlated with the analytic calculation (correlation ≈0.99, *p*-value ≤0.01). Note that the exact value of both the entropy and mutual information is different between the exact and estimated quantities. This is expected, as the matrix-based entropy estimators measure information theoretic quantities in RKHS without the need for explicit PDF but with similar properties as common information theoretic measures. The kernel width was selected by taking the median distance between all samples.

### 3.3. Novel Tensor-Based Matrix-Based Renyi Information Measures

To invoke information theoretic quantities of features produced by convolutional layers and to address the limitations discussed above, we introduce in this section our novel tensor-based approach for measuring entropy and MI in DNNs. This enables for the first time an IP analysis of large-scale CNNs.

### 3.4. Tensor Kernels for Measuring Mutual Information

The output of a convolutional layer is represented as a tensor $X_{i}\in R^{C}⊗R^{H}⊗R^{W}$ for a data point *i*. As discussed above, the matrix-based Rényi’s α-entropy cannot include tensor data without modifications. To handle the tensor-based nature of convolutional layers, we propose to utilize tensor kernels [32] to produce a kernel matrix for the output of a convolutional layer. A tensor formulation of the radial basis function (RBF) kernel can be stated as
(17)$$\kappa_{\mathrm{ten}}\left(X_{i},X_{j}\right)=e^{-\frac{1}{\sigma^{2}}{∥X_{i}-X_{j}∥}_{F}^{2}},$$
where ${∥\cdot ∥}_{F}$ denotes the Hilbert–Frobenius norm [32] and σ is the kernel width parameter. In practice, the tensor kernel in Equation (Equation 17) can be computed by reshaping the tensor into a vectorized representation while replacing the Hilbert–Frobenius norm with a Euclidean norm. We compute the MI in Equation (Equation 3) by replacing the matrix A with
(18)$$\begin{array}{cc} {\left(A_{\mathrm{ten}}\right)}_{ij} & =\frac{1}{N}\frac{{\left(K_{\mathrm{ten}}\right)}_{ij}}{\sqrt{{\left(K_{\mathrm{ten}}\right)}_{ii}{\left(K_{\mathrm{ten}}\right)}_{jj}}}\\ \; & =\frac{1}{N}\kappa_{\mathrm{ten}}\left(X_{i},X_{j}\right). \end{array}$$

While Equation (Equation 17) provides the simplest and most intuitive approach for using kernels with tensor data, it does have its limitations. Namely, a tensor kernel that simply vectorizes the tensor ignores the inter-component structures within and between the respective tensor [32]. For simple tensor data, such structures might not be present and a tensor kernel as described above can suffice; however, other tensor kernels do exist, such as for instance the matricization-based tensor kernels [32]. In this work, we have chosen the tensor kernel defined in Equation (Equation 17) for its simplicity and computational benefits, which come from the fact that the entropy and joint entropy are computed batch-wise by finding the eigenvalues of a kernel matrix, or the eigenvalues of the Hadamard product of two kernel matrices, and utilizing Equation (Equation 1). Nevertheless, exploring structure-preserving kernels can be an interesting research path in future works. In Appendix C, we have included a simple example toward this direction, where the tensor kernel described in this paper is compared to a matricization-based tensor kernel.

#### 3.4.1. Choosing the Kernel Width

With methods involving RBF kernels, the choice of the kernel width parameter, σ, is always critical. For supervised learning, one might choose this parameter by cross-validation based on validation accuracy, while in unsupervised problems, one might use a rule of thumb [33,34,35]. However, in the case of measuring MI in DNNs, the data are often high dimensional, in which case unsupervised rules of thumb often fail [34].

In this work, we choose σ based on an optimality criterion. Intuitively, one can make the following observation: a good kernel matrix should reveal the class structures present in the data. This can be accomplished by maximizing the so-called *kernel alignment* loss [36] between the kernel matrix of a given layer, $K_{\sigma }$, and the label kernel matrix, $K_{y}$. The kernel alignment loss is defined as
(19)$$A\left(K_{a},K_{b}\right)=\frac{{\langle K_{a},K_{b}\rangle }_{F}}{∥K_{a}{∥}_{F}{∥K_{b}∥}_{F}},$$
where ${∥\cdot ∥}_{F}$ and ${\langle \cdot ,\cdot \rangle }_{F}$ denote the Frobenius norm and inner product, respectively. Thus, we choose our optimal σ as
$$\sigma^{*}={arg max}_{\sigma }A\left(K_{\sigma },K_{y}\right).$$
To stabilize the σ values across mini batches, we employ an exponential moving average, such that in layer *ℓ* at iteration *t*, we have
$$\sigma_{𝓁,t}=\beta \sigma_{𝓁,t-1}+\left(1-\beta \right)\sigma_{𝓁,t}^{*},$$
where β∈[0,1] and $\sigma_{𝓁,1}=\sigma_{𝓁,1}^{*}$.

## 4. Results

We evaluate our approach by comparing it to previous results obtained on small networks by considering the MNIST dataset and a Multilayer Perceptron (MLP) architecture that was inspired by [5]. We further compare to a small CNN architecture similar to that of [4] before considering large networks, namely VGG16, and a more challenging dataset, namely CIFAR-10. Note that unless stated otherwise, we use CNN to denote the small CNN architecture. Details about the MLP and the CNN utilized in these experiments can be found in Appendix D. All MI measures were computed using Equations (Equation 2), (Equation 3) and (Equation 10) and the tensor approach described in Section 4, which amounts to setting α=1. Furthermore, the MI estimates showed in all plots are averages across multiple training runs. Code is available online (https://github.com/Wickstrom/InformationTheoryExperiment, accessed on 1 June 2023).

Since the MI is computed at the mini-batch level, a certain degree of noise is present. To smooth the MI measures, we employ a moving average approach where each sample is averaged over *k* mini-batches. For the MLP and CNN experiments, we use k=10, and for the VGG16, we use k=50. We use a batch size of 100 samples and determine the kernel width using the kernel alignment loss defined in Equation (Equation 19). For each hidden layer, we chose the kernel width that maximizes the kernel alignment loss in the range 0.1 and 10 times the mean distance between the samples in one mini-batch. Initially, we sample 75 equally spaced values for the kernel width in the given range for the MLP and CNN and 300 values for the VGG16 network. During training, we dynamically reduce the number of samples to 50 and 100, respectively, to reduce computational complexity, which is motivated by the fact that the kernel width remains relatively stable during the latter part of training (illustrated in Section 5). We chose the ranges 0.1 and 10 times the mean distance between the samples in one mini-batch to avoid the kernel width becoming too small and to ensure that we cover a wide enough range of possible values. For the input kernel width, we empirically evaluated values in the range 2–16 and found consistent results for values in the range 4–12. All our experiments were conducted with an input kernel width of 8. For the label kernel matrix, we want a kernel width that is as small as possible to approach an ideal kernel matrix while at the same time large enough to avoid numerical instabilities. For all our experiments, we use a value of 0.1 for the kernel width of the label kernel matrix.

**Comparison to previous approaches** First, we study the IP of the MLP similar to the one examined in previous works on DNN analysis using information theory [4,5]. We utilize stochastic gradient descent, a cross-entropy loss function, and repeat the experiment 5 times. Figure 1 displays the IP of the MLP with a ReLU activation function in each hidden layer. MI was measured using the training data of the MNIST dataset. A similar experiment was performed with the tanh activation function, obtaining similar results. The interested reader can find these results in Appendix E.

From Figure 1, one can clearly observe a fitting phase, where both I(T;X) and I(Y;T) increase rapidly, followed by a compression phase where I(T;X) decrease and I(Y;T) remains unchanged. In addition, note that I(Y;T) for the output layer (layer 5 in Figure 1) stabilizes at an approximate value of $\log_{2}\left(10\right)$. The following analysis shows that this is to be expected. When the network achieves approximately 100% accuracy, $I\left(Y;\hat{Y}\right)\approx S\left(Y\right)$, where $\hat{Y}$ denotes the output of the network, since *Y* and $\hat{Y}$ will be approximately identical and the MI between a variable and itself is just the entropy of the variable. The entropy of *Y* is measured using Equation (Equation 10), which requires the computation of the eigenvalues of the label kernel matrix $\frac{1}{N}K_{y}$. For the ideal case, where ${\left(K_{y}\right)}_{ij}=1$ if $y_{i}=y_{j}$ and zero otherwise, $K_{y}$ is a rank *K* matrix, where *K* is the number of classes in the data. Thus, $\frac{1}{N}K_{y}$ has *K* non-zero eigenvalues which are given by $\lambda_{k}\left(\frac{1}{N}K_{y}\right)=\frac{1}{N}\lambda_{k}\left(K_{y}\right)=\frac{N_{c_{k}}}{N}$, where $N_{c_{k}}$ is the number of datapoints in class k,k=1,2,…,K. Furthermore, if the dataset is balanced, we have $N_{c_{1}}=N_{c_{2}}=\ldots =N_{c_{K}}\equiv N_{c}$. Then, $\lambda_{k}\left(\frac{1}{N}K_{y}\right)=\frac{N_{c}}{N}=\frac{1}{K}$, which gives us the entropy measure
(20)$$\begin{array}{cc} S\left(\frac{1}{N}K_{y}\right) & =-\sum_{k=1}^{K}\lambda_{k}\left(\frac{1}{N}K_{y}\right)\log_{2}\left[\lambda_{k}\left(\frac{1}{N}K_{y}\right)\right]\\ \; & =-\sum_{k=1}^{K}\frac{1}{K}\log_{2}\left[\frac{1}{K}\right]\\ \; & =\log_{2}\left[K\right]. \end{array}$$

Next, we examine the IP of a CNN, similar to that studied by [4], with a similar experimental setup as for the MLP experiment. Figure 4 displays the IP of the CNN with a ReLU activation function in all hidden layers. A similar experiment was conducted using the tanh activation function and can be found in Appendix F. While the output layer behaves similarly to that of the MLP, the preceding layers show much less movement. In particular, no fitting phase is observed, which we hypothesize is a result of the convolutional layers being able to extract the necessary information in very few iterations. Note that the output layer is again settling at the expected value of $\log_{2}\left(10\right)$, similar to the MLP, as it also achieves close to 100% accuracy.

**Increasing DNN size** We analyze the IP of the VGG16 network on the CIFAR-10 dataset with the same experimental setup as in the previous experiments. To our knowledge, this is the first time that the full IP has been modeled for such a large-scale network. Figure 5 and Figure 6 show the IP when measuring the MI for the training dataset and the test dataset, respectively. For the training dataset, we can clearly observe the same trend as for the smaller networks, where layers experience a fitting phase during the early stages of training and a compression phase in the later stage. Note that the compression phase is less prominent for the testing dataset. Note also the difference between the final values of I(Y;T) for the output layer measured using the training and test data, which is a result of the different accuracy achieved on the training data (≈100%) and test data (≈90%). Ref. [3] claims that I(T;X)≈H(X) and I(Y;T)≈H(Y) for high-dimensional data, and they highlight particular difficulties with measuring the MI between convolutional layers and the input/output. However, this statement is dependent on their particular measure for the MI, and the results presented in Figure 5 and Figure 6 demonstrate that neither I(T;X) nor I(Y;T) is deterministic for our proposed measure. Furthermore, other measures of MI have also demonstrated that both I(T;X)≈H(X) and I(Y;T)≈H(Y) evolve during training [4,18].

Another type of widely used DNNs is residual networks [37]. While these networks typically have less parameters than the VGG16, they usually have more layers. This increase in the number of layers is enabled by skip-connections that allow data to flow through the network without loss of information. This complicates the information theoretic analysis, as the dynamics between the layers change and information do not need to decrease in between the layers. While our proposed estimator is computationally capable of handling residual networks, an extensive analysis would be required to understand the added complexity that is introduced by the lossless flow of information in these networks. We consider such an analysis as outside the scope of this paper but an interesting avenue of future research.

**Effect of early stopping** We also investigate the effect of using early stopping on the IP described above. Early stopping is a regularization technique where the validation accuracy is monitored and training is stopped if the validation accuracy does not increase for a set number of iterations, which is often referred to as the patience hyperparameter. Figure 1 displays the results of monitoring where the training would stop if the early stopping procedure was applied for different values of patience. For a patience of five iterations, the network training would stop before the compression phase takes place for several of the layers. For larger patience values, the effects of the compression phase can be observed before training is stopped. Early stopping is a procedure intended to prevent the network from overfitting, which may imply that the compression phase observed in the IP of DNNs can be related to overfitting. However, recent research on the so-called double descent phenomenon has shown that longer training might be necessary for good performance for overparameterized DNNs [38,39]. In such settings, early stopping might not be as applicable. We describe the double descent phenomenon and investigate its possible connection with the IP in Appendix H.

**Data processing inequality** The data processing inequality (DPI) is a concept in information theory which states that the amount of information cannot increase in a chain of transformations. A good information theoretic estimator should tend to uphold the DPI. DNN consists of a chain of mappings from the input through the hidden layers and to the output. One can interpret a DNN as a Markov chain [1,7] that defines an information path [1], which should satisfy the DPI [40]:(21)$$I\left(X;T_{1}\right)\geq I\left(X;T_{2}\right)\geq \ldots \geq I\left(X;T_{L}\right),$$
where *L* is the number of layers in the network. An indication of a good MI measure is that it tends to uphold the DPI. Figure 7 illustrates the mean difference in MI between two subsequent layers in the MLP and VGG16 networks. Positive numbers indicate that MI decreases, thus indicating compliance with the DPI. We observe that our measure complies with the DPI for all layers in the MLP and all except one in the VGG16 network. Furthermore, we also model the DPI for a simple MLP using the EDGE MI estimator, which has shown encouraging results on several MI estimation tasks [4]. The DPI for the EDGE estimator is shown in Figure A4 of Appendix G, which shows that the EDGE estimator also upholds the DPI. This agrees with our results with regard to the information flow in neural networks. However, a limitation of the EDGE estimator is that it is not differentiable, which can be beneficial if MI estimates are to be included in the training [41].

## 5. Kernel Width Sigma

We further evaluate our dynamic approach of finding the kernel width σ. Figure 8 shows the variation of σ in each layer for the MLP, the small CNN and the VGG16 network. We observe that the optimal kernel width for each layer (based on the criterion in Section 3.4.1) stabilizes reasonably quickly and remains relatively constant during training. This illustrates that decreasing the sampling range is a useful approach to decreasing computational complexity.

## 6. Discussion and Conclusions

In this work, we propose a novel framework for analyzing DNNs from an information theoretic perspective using a tensor-based measure of the matrix-based approach of [12]. Our experiments illustrate that the proposed approach scales to large DNNs, which allows us to provide insights into the training dynamics. We observe that the compression phase in neural network training tends to be more prominent when MI is measured on the training set and that commonly used early-stopping criteria tend to stop training before or at the onset of the compression phase. This could imply that the compression phase is linked to overfitting. However, recent research on the double descent phenomenon has shown that a longer training time might be beneficial for generalization [38,39]. In Appendix H, we perform a preliminary study that examines a potential connection between the compression phase and the recent epoch-wise double descent phenomenon. Furthermore, we showed that for our tensor-based approach, the claim that H(X)≈I(T;X) and H(Y)≈I(T;Y) does not hold. We believe that our proposed approach can provide new insight and facilitate a more theoretical understanding of DNNs.

## Figures and Tables

**Figure 1 entropy-25-00899-f001:**
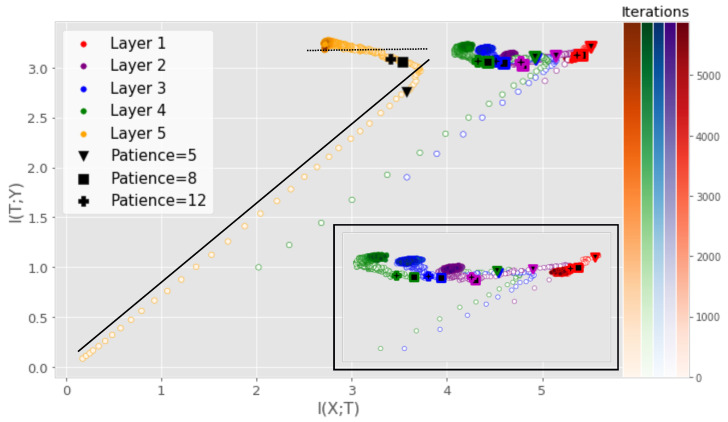
IP obtained using our proposed measure for a small DNN averaged over 5 training runs. The solid black line illustrates the fitting phase while the dotted black line illustrates the compression phase. The iterations at which early stopping would be performed assuming a given patience parameter are highlighted. Patience denotes the number of iterations that need to pass without progress on a validation set before training is stopped to avoid overfitting. For low patience values, training will stop before the compression phase. For the benefit of the reader, a magnified version of the first four layers is also displayed.

**Figure 2 entropy-25-00899-f002:**
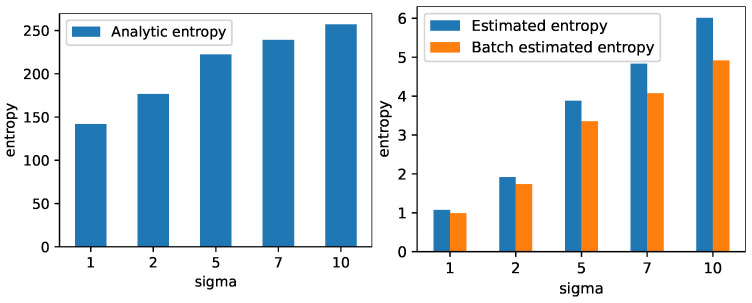
The leftmost plot shows the entropy calculated using Equation (Equation 14) of a 100-dimensional normal distribution with zero mean and an isotropic covariance matrix for different variances. The variances are given along the x-axis. The rightmost plot shows the entropy estimated using Equation (Equation 1) for the same distribution. The plots illustrated that the analytically computed entropy and the estimated quantity follow the same trend.

**Figure 3 entropy-25-00899-f003:**
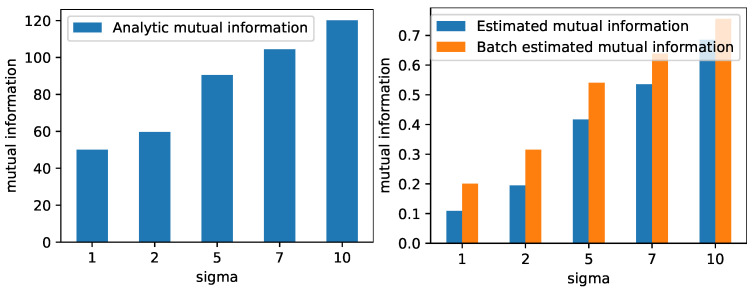
The leftmost plot shows the mutual information calculated using Equation (16) between a standard 100-dimensional normal distribution and a normal distribution with a mean vector of all ones and an isotropic covariance matrix with different variances. The variances are given along the x-axis. The rightmost plot shows the mutual information estimated using Equation (Equation 3) for the same distributions. The plots illustrated that the analytically computed mutual information and the estimated quantity follow the same trend.

**Figure 4 entropy-25-00899-f004:**
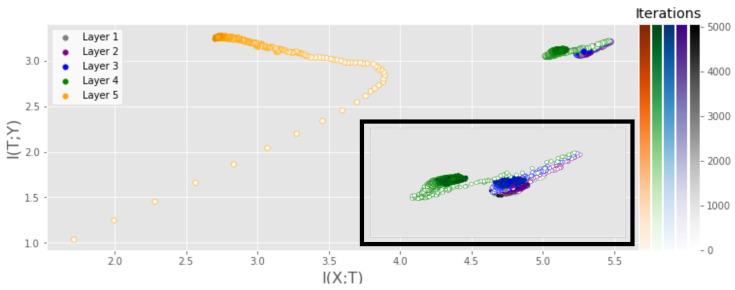
IP of a CNN consisting of three convolutional layers with 4, 8 and 12 filters and one fully connected layer with 256 neurons and a ReLU activation function in each hidden layer. MI was measured using the training data of the MNIST dataset and averaged over 5 runs.

**Figure 5 entropy-25-00899-f005:**
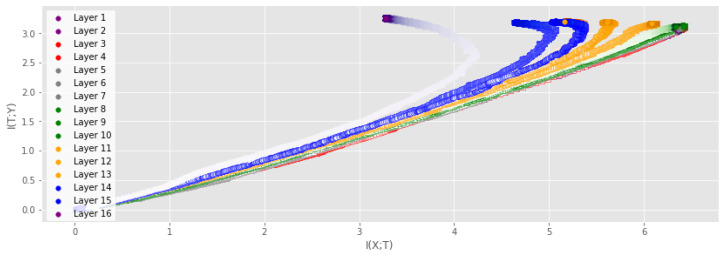
IP of the VGG16 on the CIFAR-10 dataset. MI was measured using the training data and averaged over 2 runs. Color saturation increases as training progresses. Both the fitting phase and the compression phase is clearly visible for several layers.

**Figure 6 entropy-25-00899-f006:**
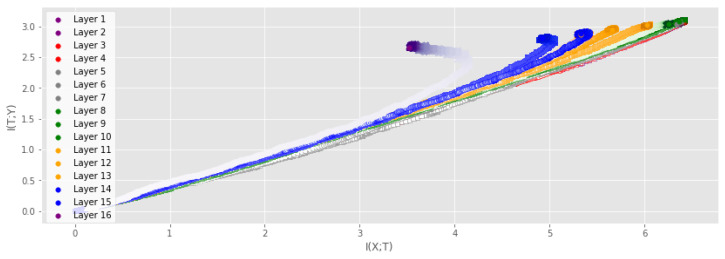
IP of the VGG16 on the CIFAR-10 dataset. MI was measured using the test data and averaged over 2 runs. Color saturation increases as training progresses. The fitting phase is clearly visible, while the compression phase can only be seen in the output layer.

**Figure 7 entropy-25-00899-f007:**
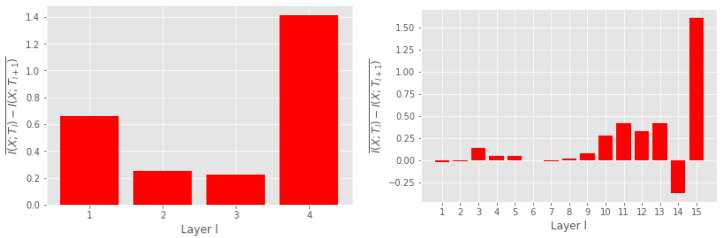
Mean difference in MI of subsequent layers *ℓ* and 𝓁+1. Positive numbers indicate compliance with the DPI. MI was measured on the MNIST training set for the MLP and on the CIFAR-10 training set for the VGG16.

**Figure 8 entropy-25-00899-f008:**
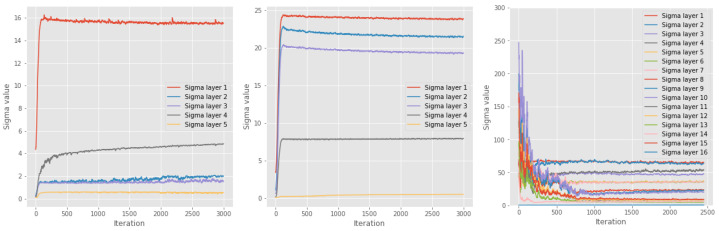
Evolution of kernel width as a function of iteration for the three networks that we considered in this work. From left to right, plots shows the kernel width for the MLP, CNN, and VGG16. The plots demonstrate how the optimal kernel width quickly stabilizes and stays relatively stable throughout the training.

## Data Availability

We used the publicly available MNIST (http://yann.lecun.com/exdb/mnist/, accessed on accessed on 1 January 2023) and CIFAR 10 (https://www.cs.toronto.edu/~kriz/cifar.html, accessed on 1 January 2023) datasets in our analysis.

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
