# Peer review of "Analysis of Deep Convolutional Neural Networks Using Tensor Kernels and Matrix-Based Entropy"

_entropy, 2023, doi:10.3390/e25060899_

Round 1

Reviewer 1 Report

This is a lovely paper; it is both a nice introduction to matrix-based entropy measures and an interesting contribution to the recent work on using information to investigate the learning dynamics of neural networks. This is unlikely to be of high interest because there are so many different ways of looking at, and interpreting, these dynamics, but it certainly is of interest. It is also very well written. I have no hesitation in recommending publication.

It is only a small remark, but they not that Shannon's entropy suffers the curse of dimensionality, I don't believe that's the case, Shannon's entropy itself does not suffer this curse, though many estimators do, there are metric based estimators of Shannon's entropy, such as

Calculating the mutual information between two spike trains
Conor Houghton,
Neural Computation (2019) 31:330-343

and indeed this has a kernel version

A kernel-based calculation of information on a metric space.
R. Joshua Tobin and Conor J. Houghton,
Entropy 15 (2013) 4540-4552.

I wonder what the relationship between the this and the metric based version is! However, this is only a passing remark, not a suggestion for any change to the manuscript, which I think should be published as is.

Author Response

Thank you for your time and effort in reviewing our manuscript. We are delighted to hear that you found our work interesting and worthy of publication.

You are correct in that the Shannon entropy itself does not suffer from the curse of dimensionality. Rather, the common estimators used to approximate the distribution of the data often suffer from the curse of dimensionality. This could have been stated more clear, so we have changed Section 3.1.2 in our revised manuscript from:

"Not all measures of entropy have the same properties. The ones developed for Shannon suffer from the curse of dimensionality"

to

"Not all measures of entropy have the same properties. Many of the estimators used and developed for Shannon suffer from the curse of dimensionality"

With regards to your remark, there are definitely some interesting connections with the metric-based versions. But the important difference is that the matrix-based Renyi estimator is not measured in probability space, but quantify information from the eigenspectrum (taking inspiration from quantum information theory).

Reviewer 2 Report

I thank the authors for their interesting manuscript. I find the topic of information plane analyses extremely interesting and controversial, and any additional piece of evidence is welcome. Unfortunately, the manuscript is presently not ready for publication, at least in my opinion. I find that there is some lack of clarity,and that some of the experiments are insufficient to justify the drawn conclusions. Below are some comments that may help the authors to improve the manuscript.

- Most problematic is the synthetic example in Section 4.2, which may reveal a problem with the MI estimate. Namely, in Fig. 3 the authors show that the MI between two RVs increases as the overlap of the corresponding distributions increases. This is counter-intuitive. MI is a measure of statistical dependence, not one of the similarity of distributions. In other words, even if the distribution of both RVs are the same (i.e., the point clouds perfectly overlap), the mutual information may still be zero if the RVs are drawn independently. The authors do not specify if the draws from the two Gaussian distributions are independent -- but in any case, an increase of MI with increasing overlap of distributions is not what one would expect to see for a proper measure of mutual information.
- It would be good to explicitly state the value of alpha for the experiments. The authors refer to equation (10) from which I conclude that alpha=1 was used. However, stating this explicitly may be useful.
- In the light of this, the mathematical result in (13) appears to be disconnected from the rest of the paper since it applies for alpha=2. It would be good to expand on this.
- The authors average MI over multiple mini-batches and (apparently) only report the average, but not the standard deviation. Further, the authors e.g. repeat the experiment with the MLP 5 times (line 291). I assume that the IP shows the average, but this should be clearly stated.
- The experimental evidence for the connection between epoch-wise double descent and compression is insufficient. Experiments have been performed for only a single network architecture, averaged over only three runs. Further, the claimed connection between compression and double descent is misleading: compression is usually defined via the term I(X;H), while the authors claim a connection of double descent with I(X;Y^). However, since I(X;Y^) is directly connected to the network output Y^ (if I understood it correctly), a connection to the test loss is not surprising.

Based on this, I tend to rejecting the manuscript. Giving the authors the benefit of doubt, I now suggest a major revision, hoping that my main concerns can either be resolved or at least addressed in the paper.

Minor comments:
- The theorem/proposition environments seem to be broken, hence so are the references to them.

Author Response

Thank you for your thorough evaluation of our manuscript. We highly appreciate your comments and we believe they have improved our manuscript. You can find our point-by-point response in the attached document. We hope to have provided satisfactory answers.

Round 2

Reviewer 2 Report

Dear authors, thanks for your classifications. I really appreciate the effort you put into revising the manufacturer and updating the paper. Unfortunately, my main concern (number 1) still stands: your new equation (15) in the response letter is not the formula for mutual information, but for the Kullback-Leibler divergence between two Gaussian distributions. KLD measures something completely different, though: N1 and N0 may be independent (true MI is zero), but your quantity will be large if the distributions are similar. I have the feeling that this impacts the main statements of the papers and that a more substantial revision is required. I hence suggest to revise and resubmit the paper at a later time.

Of course, I may have misunderstood some aspects, in which case I am looking forward to further discussions.

Author Response

Dear reviewer,

Thank you for your appreciation of our work and your feedback. You are completely correct, our update on your main concern was incorrect, and did not improve on the prior version. Please see the attached document where we have corrected this error and provided a detailed and comprehensive reply to your comment. We are very grateful for your vigilance in identifying this mistake, as it would certainly cause confusion for readers and degrade the quality of our manuscript. We believe that the updated version has provided clarity to the synthetic experiment, and hope it contributes to regaining your trust after the error in the previous versions. 

Round 3

Reviewer 2 Report

I thank the authors for the clarification. The new experiment is much more in line with what one would expect and the results seem reasonable. I still have minor reservations about the validity of the estimator (also reinforced by the poor performance of the estimator in https://arxiv.org/abs/2301.08164) -- but now the results seem consistent. I therefore recommend acceptance of the article, with the minor wish to see the source code published with which the results were obtained.

Also, equation (15) is not standard and can be removed without replacement. The rest of the explanations is sufficient (and the experiment is described very well!).

Again, thanks for the continued discussion. I am happy that the error in the experiment was resolved, I hope it was not too much of an effort.

I am not qualified to assess the quality of English in this paper.

Author Response

Dear reviewer,

Thank you for the encouraging feedback and interesting discussion throughout this review process. We are also happy that the error was resolved, and the effort was certainly worth it.

With regards to your requests, we have not included a link to a GitHub repository (https://github.com/Wickstrom/InformationTheoryExperiment) that contains code used in the paper, and we have removed Equation 15. As you point out, it does not actually add anything to the experiment.